# Increased serum caspase-1 in adult-onset Still's disease

**Haruki Matsumoto[1], Shuhei Yoshida[1], Tomohiro Koga[2], Yuya Fujita[1], Yuya Sumichika[1], Kenji Saito[1], Jumpei Temmoku[1], Tomoyuki Asano[1], Shuzo Sato[1], Masashi Mizokami[3], Masaya Sugiyama[4], Kiyoshi Migita[1,5]***

1 Department of Rheumatology, Fukushima Medical University School of Medicine, Fukushima, Japan,
2 Department of Immunology and Rheumatology, Division of Advanced Preventive Medical Sciences, Nagasaki University Graduate School of Biomedical Sciences, Nagasaki, Japan, 3 Genome Medical Sciences Project, National Center for Global Health and Medicine, Ichikawa, Chiba, Japan, 4 Department of Viral Pathogenesis and Controls, National Center for Global Health and Medicine, Ichikawa, Chiba, Japan, 5 Department of Rheumatology, St Francisco Hospital, Nagasaki, Japan

* migita@fmu.ac.jp

## Abstract

### Background

Caspase-1 is a crucial component in the inflammasome activation cascade. This study evaluated the potential of serum caspase-1 level as an inflammatory biomarker in patients with adult-onset Still's disease (AOSD).

### Methods

The study included 51 consecutive patients diagnosed with AOSD based on the Yamaguchi criteria, 66 patients with rheumatoid arthritis (RA) as disease control, and 36 healthy controls (HCs). Serum caspase-1 concentrations were measured using enzyme-linked immunosorbent assay. The serum 69 cytokine levels were analyzed using a multisuspension cytokine array in patients with AOSD, and a cluster analysis of each cytokine was performed to determine specific molecular networks.

### Results

Patients with AOSD had significantly increased serum caspase-1 levels versus patients with RA ($p < 0.001$) and HCs ($p < 0.001$). Additionally, serum caspase-1 demonstrated significant positive correlations with AOSD disease activity score (Pouchot score, r = 0.59, $p < 0.001$) and serum ferritin (r = 0.54, $p < 0.001$). Furthermore, among patients with AOSD, significant correlations existed between serum caspase-1 and inflammatory cytokines, including interleukin-18. Immunoblot analysis detected the cleaved form of caspase-1 (p20) in the serum of untreated patients with AOSD, not in those from patients with inactive AOSD receiving immunosuppressive treatments.

**Data Availability Statement:** The data underlying the results presented in the study are available from Sachiyo Kanno as the non-author institutional point of contact(sa-kanno@fmu.ac.jp).

**Funding:** The study was supported by the Japan Grant-in-Aid for Scientific Research (20K08777). The funders had no role in study design, data collection and analysis, decision to publish, or preparation of the manuscript.

**Competing interests:** No competing interests exist.

## Conclusions

Caspase-1 is a useful biomarker for AOSD diagnosis and monitoring. Caspase-1 activation could be correlated with the inflammatory component of AOSD, specifically through proinflammatory cytokine induction via inflammasome activation cascades.

## Introduction

Adult-onset Still's disease (AOSD) is a systemic autoinflammatory disease characterized by a broad spectrum of clinical manifestations [1]. Its pathogenesis is not fully understood, but inflammasome activation may be involved [2]. Nucleotide-binding oligomerization domain, leucine-rich repeat, and pyrin domain 3 (NLRP3) inflammasome activation trigger caspase-1 activation, which converts pro-interleukin (IL)-1β and pro-IL-18 into their mature forms [3]. Excessive inflammasome activation causes IL-1β and IL-18 overproduction in autoinflammatory disorders [4]. Hence, the autoinflammatory processes in AOSD involved IL-1β and IL-18 produced by innate immune cells through inflammasome activation [5].

Caspase-1 is the prime member of the inflammatory caspases, and it activates pro-IL-1β or pro-IL-18 [6]. Activated caspase-1 may result in the processing of IL-18 and IL-1β into mature forms, which play an important role in cytokine cascades in AOSD, during the inflammasome activation processes [7]. Previous studies revealed that serum caspase-1 was elevated in patients with inflammatory disorders, including neurological or hepatic diseases [8,9]. However, the use of caspase-1 as a serum biomarker for autoinflammatory disorders remains unknown, and the role of caspase-1 in rheumatic diseases is still unclear. This study investigated the clinical relevance of caspase-1 as an autoinflammatory biomarker in AOSD, considering the importance of inflammasomes in its pathogenesis. We analyzed serum caspase-1 levels in patients with AOSD versus those with rheumatic arthritis (RA) and healthy controls (HCs). Additionally, we investigated the clinical correlations of caspase-1 with inflammatory cytokines and disease parameters in patients with AOSD.

## Materials and methods

### Patients and study design

This study enrolled 51 untreated patients with AOSD who were treated at Fukushima Medical University Hospital Department of Rheumatology from January 1995 to April 2020. All patients included had to be ≥17 years old to be diagnosed with AOSD following Yamaguchi's diagnostic criteria [10] after excluding those with infectious, neoplastic, and autoimmune disorders.

We additionally collected serum samples from 18 patients with AOSD treated during remission to investigate the longitudinal changes. As controls, 36 HCs (12 men and 24 women, median age: 39 years, interquartile range [IQR]: 32–45 years) were included. HCs lacked chronic medical diseases or conditions and did not take prescription medications or over-the-counter medications within 7 days. Additional independent sets consisting of 66 patients with rheumatoid arthritis (RA) were used to identify the specificity of the values of caspase-1 in patients with AOSD. Patients diagnosed with RA in Fukushima Medical University Hospital were randomly selected. This study was conducted following the principles of the Declaration of Helsinki and approved by the institutional review boards of Fukushima Medical University (No. 2021–290; approval date: April 16, 2023) and the National Center for Global

Health and Medicine (NCGM-G-003472; approval date: April 19, 2023). The institutional Review Board waived the requirement for written informed consent from participants due to the non-interventional design of the retrospective study. All data were accessed for this research from October 2023 to December 2023.

## Enzyme-linked immunosorbent assay (ELISA) methods

Serum caspase-1 concentrations were measured using an ELISA kit (R&D Systems, Minneapolis, MN, USA) following the manufacturer's instructions. The mean of minimum detectable dose (MDD) in this study was <0.20 pg/mL.

## Chemokine and cytokine measurements

Bio-Plex three-dimensional (3D) system (Bio-Rad, Hercules, CA) was used to determine the multiplex assay of humoral factors following the manufacturers' instructions. Briefly, Bio-Rad 3D and Bio-Plex Pro Wash Station equipped with a magnetic manifold was used to assay serum samples. Cytokine levels in assayed samples were derived from the standards run for each assay plate and reported as serum cytokines/chemokines in pg/mL. The Bio-Plex 3D system (Bio-Rad, Hercules, CA) and the HISCL-5000 (Sysmex Corp., Kobe, Japan) were used to quantify 67 humoral factors following the manufacturers' instructions.

## Immunoblot analysis

Human serum samples were diluted 10-fold with PBS. We separated this diluted human serum sample (total serum amounts: 2 μl) plus protein loading buffer of 5 μl by NuPAGE 3%–8% Tris-acetate gel electrophoresis (Invitrogen, Carlsbad, CA, USA). Proteins were electrophoretically transferred onto an Invitrogen polyvinylidene fluoride membrane and incubated overnight at 4˚C with a blocking solution [5% non-fat milk in Tris-buffered saline with 0.05% Tween 20 (TTBS)]. The blocked membrane was incubated with rabbit anti-human caspase-1 (full-length p48, D7F10, Cell Signaling Technology, Boston, MA, USA) antibody or cleaved caspase-1 antibody (p20 D57A2, Cell Signaling) at 1:1000 dilution with 1% non-fat milk in TTBS for 1 h at room temperature and then washed five times with TTBS buffer for 10 min each time at room temperature with constant shaking. The membrane was then incubated with horseradish peroxidase-conjugated second antibody (1:3000 dilution; Santa Cruz Biotechnology, Dallas, TX, USA) for 1 h at room temperature and washed five times with TTBS buffer for 10 min each time at room temperature with constant shaking. An electrochemiluminescence western blotting kit (Amersham, Little Chalfont, UK) was used for immunodetection analysis. Images of the developed film were scanned using a LAS-3000 image analyzer (Fujifilm, Tokyo, Japan). As an internal control, anti-human transferrin (1:7000 dilution; mouse, 661771-1-Ig, Proteintech), was used to detect transferrin using the above-described process.

## Statistical analysis

Data are presented as medians and IQR for continuous variables and as frequencies and percentages for qualitative variables. Spearman's rank correlation test was used to calculate correlations between pairs of serum markers. We performed the multivariate classification algorithm of random forest analysis using the R package RandomForest (http://cran.r-project.org/web/packages/randomForest/) version 4.6.12 software, as previously described, to rank the cytokine levels [11]. Mann-Whitney's U test was used to compare group differences. Wilcoxon signed-rank tests were used to evaluate changes pre- and post-treatment. R software (version 4.3.1) and Statistical Package for the Social Sciences Statistics software (version 29.0;

IBM Corp., Armonk, NY, USA) were used for statistical analyses. The receiver operating characteristic (ROC) curve was used to identify the cut-off value for caspase-1 in distinguishing between AOSD and RA. All reported $p$-values were two-sided, and a $P$-value of $<0.05$ was considered statistically significant. Bonferroni's correction was applied to the multiplex analysis.

## Results

### Demographic data of patients with AOSD

This study included 51 patients with AOSD (88.2% female; median age: 40 years, IQR: 28–56), whose serum samples were collected in the active state. The laboratory parameters include a complete blood count, liver function tests, C-reactive protein (CRP), and ferritin. Table 1 summarizes the baseline characteristics and laboratory data. The principal clinical symptoms included a high spiking fever (82.4%), skin rash (68.7%), arthralgia (57.8%), sore throat (41.2%), and splenomegaly (39.2%). Patients with AOSD demonstrated elevated median levels of biological markers that represent disease activity, such as CRP (median: 6.8 mg/dl, IQR: 2.9–10.9) and ferritin (median: 1,159 pg/ml, IQR: 310–3,887). The maximum prednisolone dose during the active phase, the number of patients who received steroid pulse, and concomitant immunosuppressive drugs are described.

### Demographic data of patients with RA and HCs

The study included 66 patients with RA and 36 HCs. Of the patient with RA, 39 (59.0%) were female, and their median age was 67 years (IQR: 62–74 years). Additionally, 8 (12.1%) patients with RA were untreated. Most of the patients with RA were taking disease-modifying antirheumatic drugs, mostly methotrexate (26/66, 39.4%), and biologics (17/66, 25.8%). The median DAS28-CRP was 2.7 (IQR: 2.2–3.6), and the median DAS28-ESR was 3.4 (IQR: 3.0–4.0). Of the HCs, 24 were women, and the average age was 41 years [IQR]; 28–52 years. They had no medical history or ongoing medications, and recent physical examinations revealed no abnormalities. S1 Table presents the demographic data of patients with RA and HCs.

### Serum caspase-1 levels in patients with AOSD

Serum caspase-1 levels were determined using ELISA in patients with AOSD, those with RA, and HCs. Patients with AOSD demonstrated significantly higher caspase-1 levels [median: 419.33 ng/ml, IQR (79.19–555.84)] compared to those with RA (6.98 ng/ml, $p < 0.001$) and HCs (5.85 ng/ml, $p < 0.001$) (Fig 1). We analyzed the cut-off value for caspase-1 in distinguishing between AOSD and RA by obtaining the ROC curve, which revealed 26.9 ng/mL (sensitivity of 100.0% and specificity of 93.9%). The area under the ROC curve was 0.987 (Fig 2).

Moreover, we evaluated the correlations of serum caspase-1 with various clinical parameters in patients with AOSD. Serum caspase-1 did not correlate with leucocyte counts or transaminases, but it was significantly correlated with serum ferritin and the Pouchet score (Fig 3), which correspond to the disease activity of AOSD [12]. However, no correlation was observed between CRP and caspase-1 (r = 0.143, $p = 0.356$) (data was not shown in the figure). Additionally, the comparison of caspase-1 between MAS and non-MAS revealed no significant difference ($p = 0.189$).

We included 18 patients with 2 longitudinal samples (at least 1 month apart) to investigate the longitudinal changes in caspase-1. The longitudinal study followed 8 patients with active AOSD until they became inactive, and then they were resampled, revealing 6 non-MAS and 2 cases were MAS cases. Serum caspase-1 levels significantly decreased (Fig 4) alongside ferritin and Pouchot's score after immunosuppressive treatments.

**Table 1. Demographic and clinical characteristics of patients with AOSD.**

| Variables | n = 51 |
|---|---|
| Female, n (%) | 45 (88.2) |
| Age at AOSD diagnosis (years), median (IQR) | 40 (28–56) |
| Ferritin at diagnosis (ng/mL), median (IQR) | 1,159 (310–3,887) |
| CRP at diagnosis (mg/dL), median (IQR) | 6.8 (2.9–10.9) |
| WBC at diagnosis (/μL), median (IQR) | 9,900 (7,300–13,800) |
| AST at diagnosis (U/L), median (IQR) | 87 (54–187) |
| ALT at diagnosis (U/L), median (IQR) | 35 (27–94) |
| Pouchot's score | 3 (2–5) |
| Fever, n (%) | 42 (82.4) |
| Skin rash, n (%) | 35 (68.7) |
| Pleuritis, n (%) | 7 (13.7) |
| Pneumonia, n (%) | 4 (7.8) |
| Pericarditis, n (%) | 2 (3.9) |
| Increase in AST/ALT, n (%) | 24 (47.0) |
| Splenomegaly, n (%) | 20 (39.2) |
| Lymphadenopathy, n (%) | 14 (27.5) |
| Leukocytosis, n (%) | 11 (21.6) |
| Sore throat, n (%) | 21 (41.2) |
| Myalgia, n (%) | 8 (15.7) |
| Abdominal pain, n (%) | 2 (3.9) |
| Monocyclic type, n (%) | 18 (35.3) |
| Polycyclic systemic type, n (%) | 30 (58.8) |
| Chronic articular type, n (%) | 1 (1.9) |
| MAS, n (%) | 8 (15.7) |
| Maximum prednisolone dose (mg/day), median (IQR) | 42.5 (40–60) |
| Intravenous methylprednisolone, n (%) | 27 (52.9) |
| Concomitant other immunosuppression | |
| Methotrexate, n (%) | 17 (33.3) |
| Azathioprine, n (%) | 2 (3.9) |
| Cyclosporine, n (%) | 19 (37.3) |
| Cyclophosphamide, n (%) | 2 (3.9) |
| Infliximab, n (%) | 2 (3.9) |
| Etanercept, n (%) | 1 (2.0) |
| Tocilizumab, n (%) | 2 (3.9) |

All data are expressed as median (IQR), or numbers (percentages).

IQR: Interquartile range, AOSD: Adult-onset Still's disease, CRP: C-reactive protein, WBC: White blood cell, AST: Aspartate aminotransferase, ALT: Alanine aminotransferase, MAS: Macrophage activation syndrome.

## Correlation between serum caspase-1 and inflammatory cytokines and chemokines in AOSD

A total of 71 cytokines (6Ckine/ chemokine (C-C motif) ligand (CCL) 21, BCA-1/CXCL13, CTACK/CCL27, ENA-78/CXCL5, eotaxin-2/CCL24, eotaxin-3/CCL26, eotaxin/CCL11, fractalk-ine/CX3CL1, GCP-2/CXCL6, GM-CSF, Gro-α/CXCL1, Gro-β/CXCL2, interferon (IFN)-γ, I-309/CCL1, interferon-inducible T cell alpha chemoattractant (I-TAC)/CXCL11, IL-10, IL-12/IL-23 p40, IL-16, IL-1β, IL-2, IL-4, IL-6, IL-8/CXCL8, IP-10/CXCL10, MCP-1/CCL2, MCP-2/CCL8, MCP-3/CCL7, MCP-4/CCL13, MDC/CCL22, macrophage migration inhibitory factor (MIF),

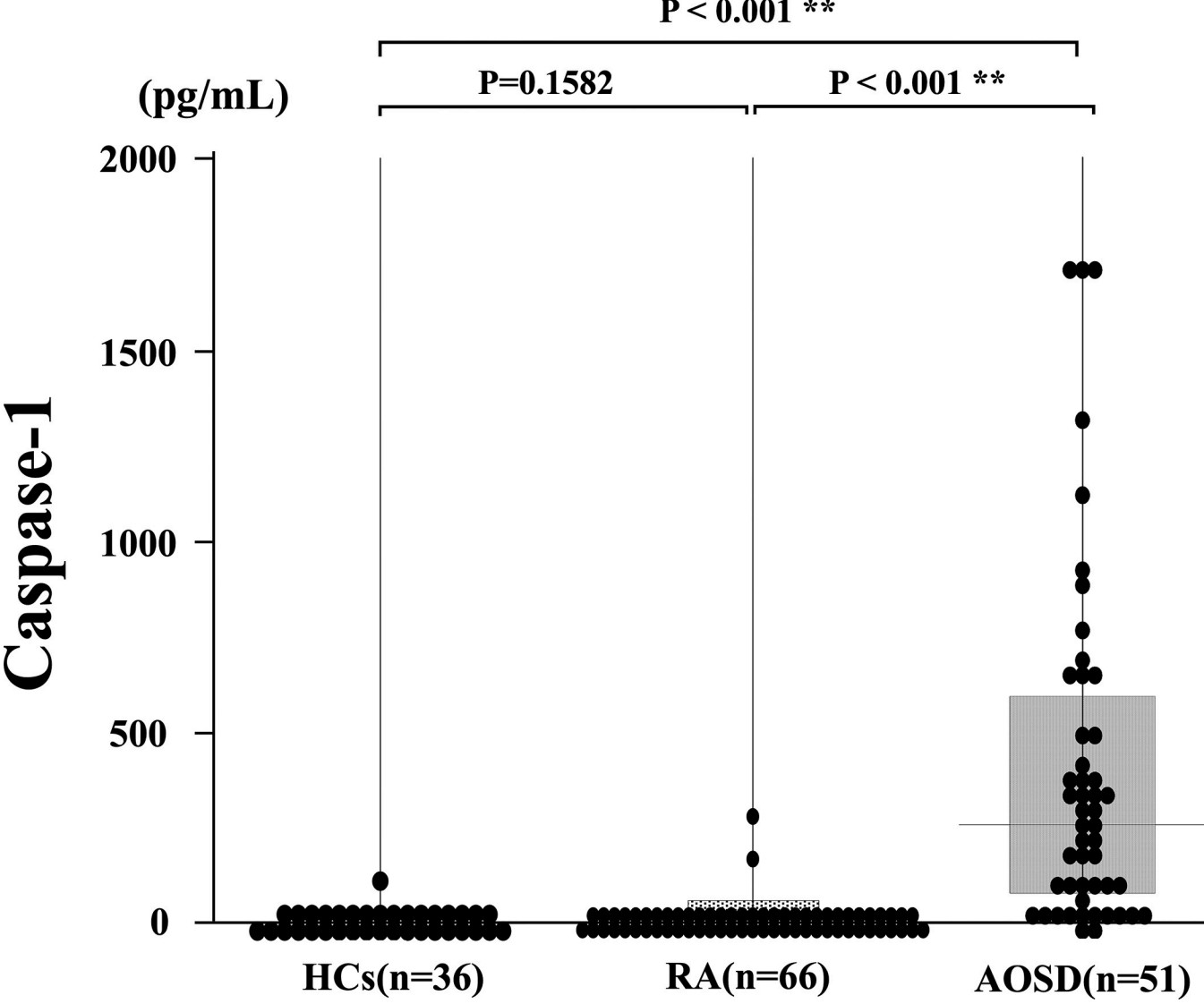

**Fig 1. Serum caspase-1 levels in patients with AOSD.** Serum caspase-1 levels in patients with AOSD (n = 51) were significantly higher compared to those with RA (n = 66) or healthy controls (n = 36). Comparisons of serum caspase-1 levels by the Kruskal–Wallis test are significantly different between patients with AOSD and those with RA as well as HCs. The AOSD group demonstrated significantly higher caspase-1 levels than both HCs and RA groups ($p < 0.001$). Post hoc pairwise analyses between three groups were analyzed by the Games–Howell test.

MIG/CXCL9, macrophage inflammatory protein (MIP)-1α/CCL3, MIP-1δ/CCL15, MIP-3α/CCL20, MIP-3β/CCL19, MPIF-1/CCL23, SCYB16/CXCL16, SDF-1α+β/CXCL12, TARC/CCL17, thymus expressed chemokine (TECK)/CCL25, tumor necrosis factor (TNF)-α, basic fibroblast growth factor (FGF), G-CSF, HGF, IFN-α2, IL-12(p40), IL-12(p70), IL-13, IL-15, IL-17, IL-18, IL-1α, IL-1Ra, IL-2Ra, IL-3, IL-5, IL-7, IL-9, leukemia inhibitory factor (LIF), macrophage colony-stimulating factor (M-CSF), MIP-1β, PDGF-BB, RANTES, SCF, SCGF-β, TNF-β, TRAIL, VEGF, β-NGF, IFN-λ3, and TARC) were measured in patients with AOSD and analyzed for their correlations with caspase-1. A heat map was used to analyze cytokine expression profiles (Fig 5). Those clustered in group 1 have positive correlations with caspase-1: eotaxin, HGF, IFN-α2, IFN-g, IL-12p40, IL-13, IL-15, IL-18, IL-2, IL-2Ra, IL-3, IL-5, IL-6, and IP-10. Among these cytokines, 17 cytokines (MIP-3α/CCL20, IL-3, MIP-3β/CCl19, LIF, IL-2Ra, I-TAC/CXCL11, Basic FGF,

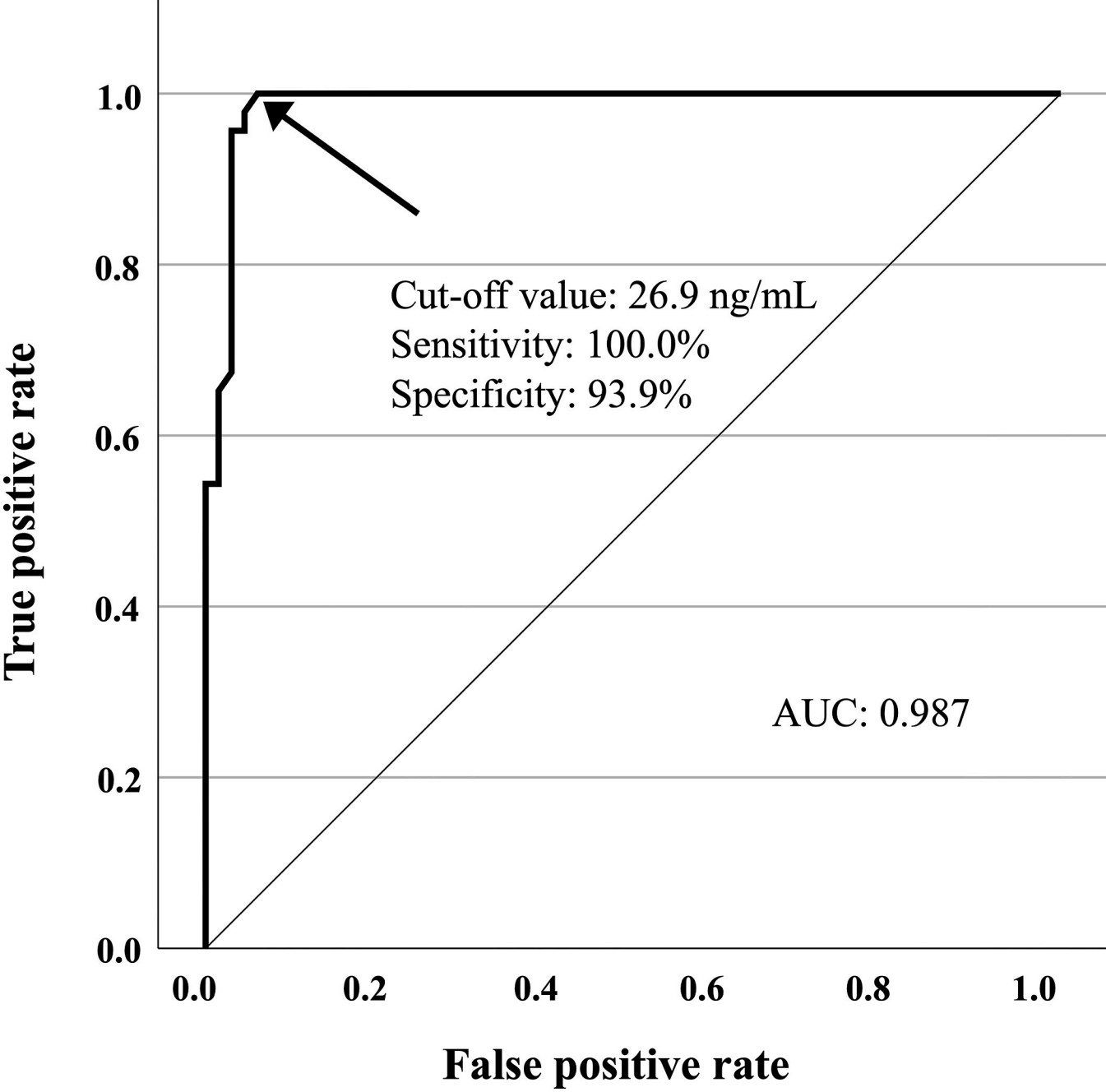

**Fig 2. The ROC curve of caspase-1 distinguishes between AOSD and RA.** The ROC curve of caspase-1 was 26.9 ng/mL for distinguishing between AOSD and RA (sensitivity of 100.0% and specificity of 93.9%). The area under the ROC curve was 0.987.

6Ckine/CCL21, MIF, IL-2, IL-1Ra, TECK/CCL25, CCL1, M-CSF, TNF-α, G-CSF, IL-18) were positively correlated with caspase-1 after Bonferroni's correction (Table 2).

### Detection of circulating cleaved caspase-1 in untreated patients with AOSD

Serum samples from patients with AOSD patients underwent immunoblot analysis using an anti-full-length caspase-1 antibody to detect the circulating caspase-1. Single bands with

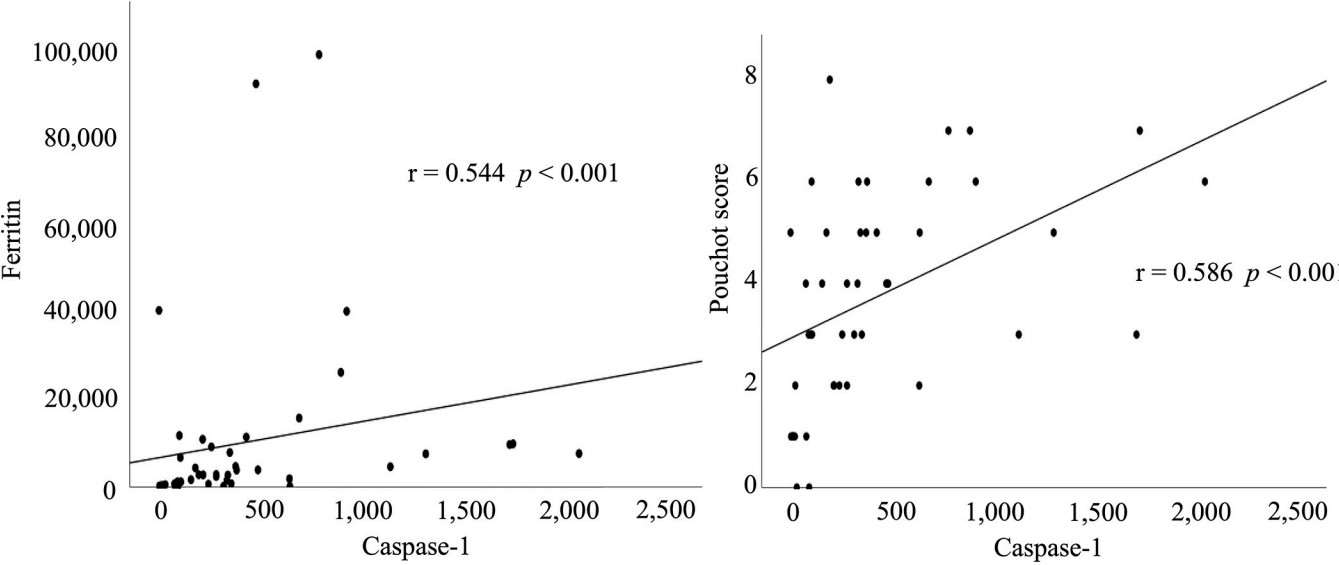

**Fig 3. Correlation between serum caspase-1 and Ferritin levels or systemic score (Pouchot score) in patients with AOSD.** Significant correlations were found between serum caspase-1 and ferritin levels or systemic scores in patients with AOSD.

molecular weights (45 kDa) were detected in the sera from patients with AOSD as well as in those from HCs, indicating that they originated from the heavy chain of serum-containing IgG. However, the full-length caspase-1 band (p48) was not detected in serum from patients with AOSD (Data not shown). Caspase-1 activation occurs during its release, and it is not an intracellular event in response to exogenous inflammasome activation [13]. Extracellular caspase-1 is induced during its cleavage step in pro-IL-1β processing [14]. Therefore, immunoblot analysis using anti-cleaved caspase-1 antibody (p20) was conducted. Serum IgG fragmentation into IgG light chains (25 kDa, respectively) was detected by a secondary antibody used for western blot (Fig 6A, upper gel). Sera were pre-absorbed using protein G beads and subjected to anti-cleaved caspase-1 western blot to avoid these interfering IgG-derived bands. Single-cleaved caspase-1 bands (20 kDa) were exclusively detected in the sera from untreated patients with AOSD, not in HCs (Fig 6A, lower gel). However, cleaved caspase-1 was detected in the sera of untreated patients with AOSD, which was diminished in the paired sera after treatment (Fig 6A, lower gel). We confirmed that serum transferrin was detected, as the internal control, to the same extent in untreated and treated patients with AOSD (Fig 6B).

## Discussion

Inflammasomes are multiprotein complexes that serve as pattern-recognition receptors in the immune system [15]. Inflammasomes can be activated by recruiting and activating caspase-1 upon sensing pathogen-associated molecular patterns [16]. Activated caspase-1 converts IL-1β and IL-18 precursors into their mature forms [3], thereby playing an important role in the pathogenesis of autoinflammatory diseases [17]. In this study, ELISA analysis revealed significantly higher caspase-1 levels in patients with active AOSD, and the elevated caspase-1 levels were correlated with serum ferritin and the Pouchet score, which are the systemic activation markers for AOSD. Furthermore, serum caspase-1 was correlated with serum IL-18, which is a crucial mediator matured by inflammasomes and is presumed as an AOSD-specific biomarker. The correlation of caspase-1 with the disease activity scores for AOSD revealed that the elevated caspase-1 levels may reflect the inflammasome activation status, which is

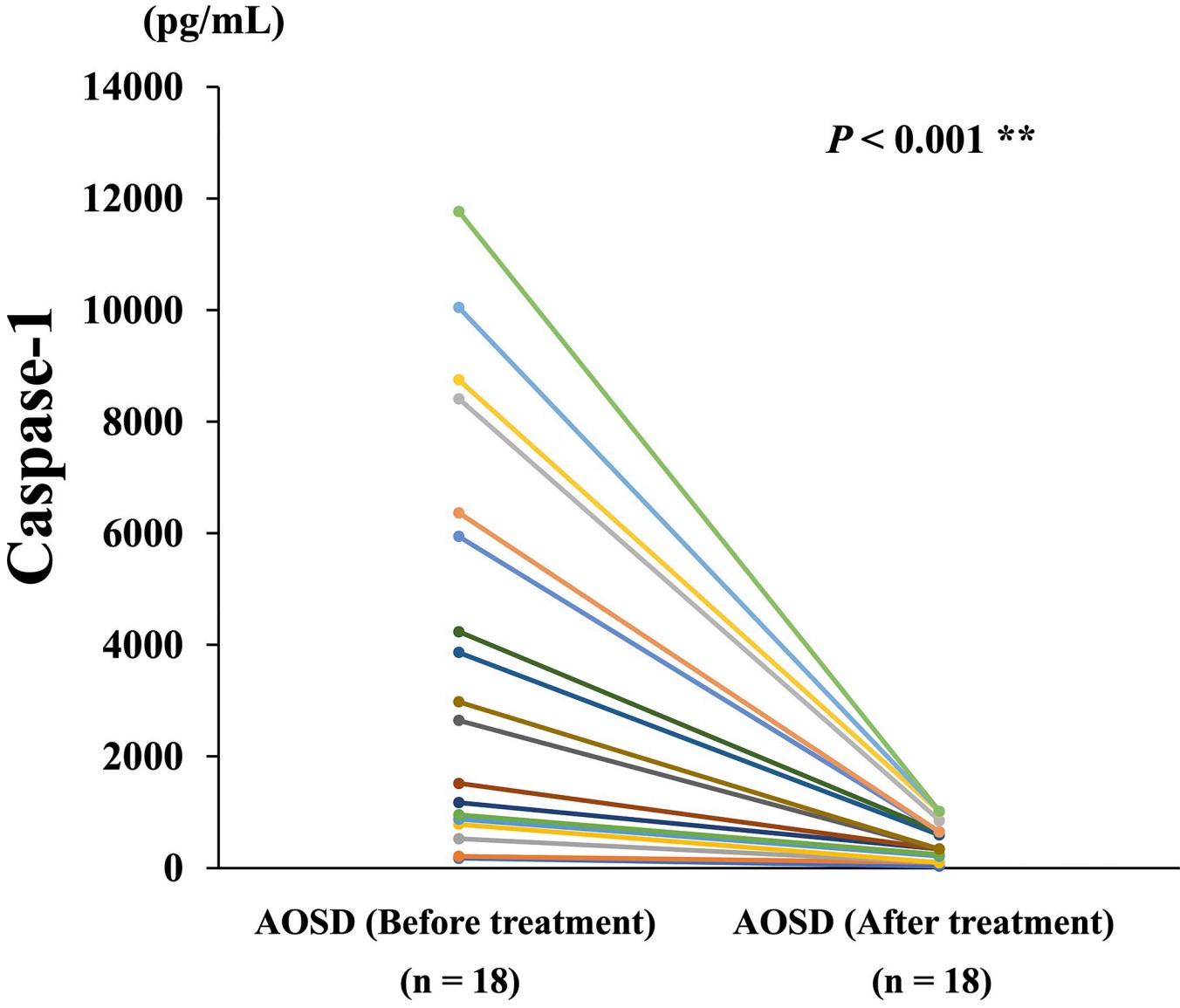

**Fig 4. Comparison of serum caspase-1 levels before and after treatment.** Elevated serum caspase-1 levels in untreated patients with AOSD were significantly decreased after immunosuppressive treatments.

implicated in the pathogenesis of AOSD. Altogether, caspase-1 serves as a novel biomarker for assessing and monitoring AOSD disease activity.

Inflammasomes are composed of three main components: NLRP3, ASC (an adaptor protein termed as apoptosis-associated speck-like protein containing a caspase recruitment domain), and pro-caspase-1, which mounts an inflammatory response against cellular damage [18]. Inflammasome activation, in turn, promotes caspase-1 activation [19]. Activated caspase-1 cannot be endogenously detected in the cytosol of innate immune cells upon inflammasome activation [20]. However, the cleaved caspase-1 subunit was detected, indicating that mature caspase-1 is released along with processed IL-1β and IL-18 during the inflammasome activation cascade and serves as a marker for inflammasome activation [21]. Activated caspase-1 triggers Gasdermin D (GSDMD) and pro-IL−1β or pro-IL-18 cleavage [22]. The cleaved GSDMD promotes pore formation in the plasma membrane and further facilitates the release of these inflammatory

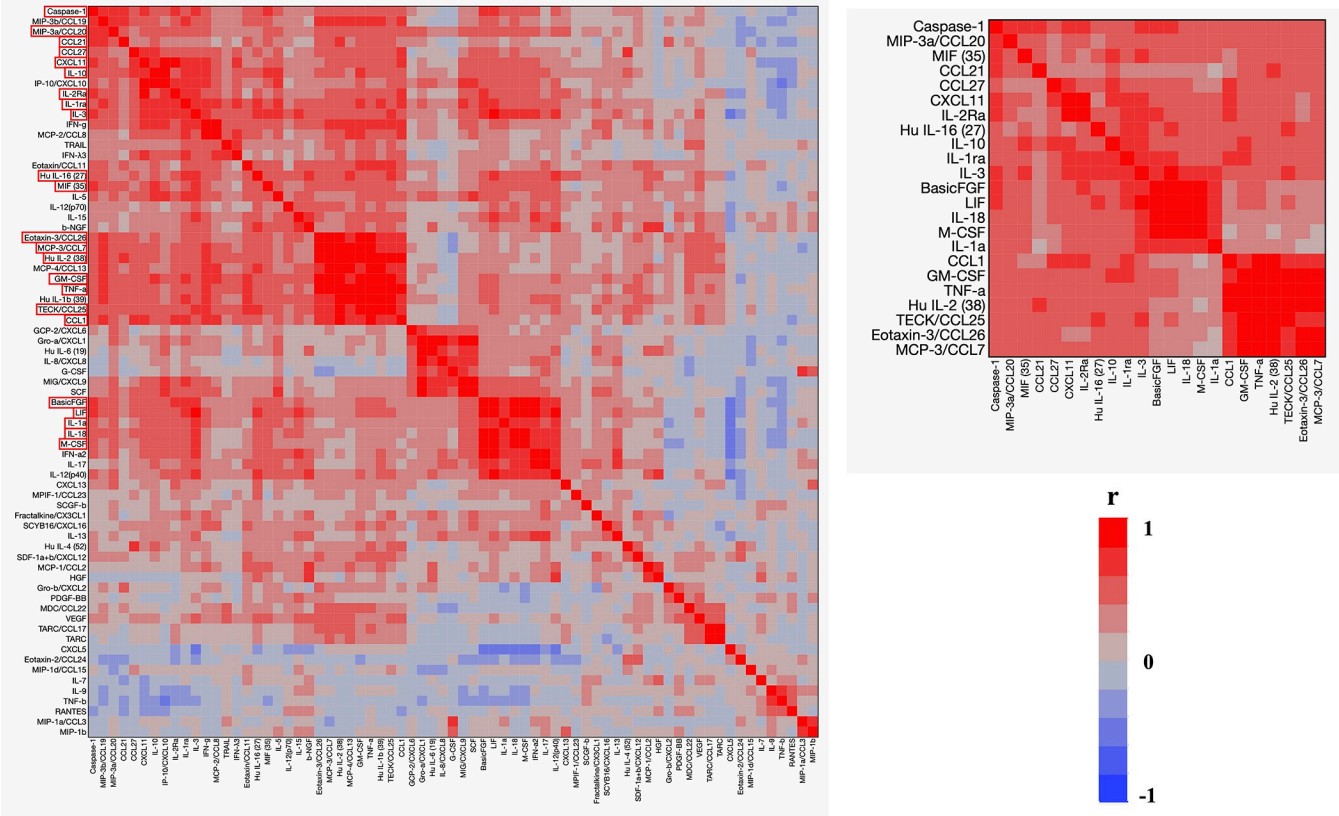

**Fig 5. Cytokine networks in patients with AOSD.** Hierarchical clustering with a Pearson correlation heatmap of serum cytokine levels and caspase-1 among patients with AOSD.

**Table 2. Correlations between serum cytokines and caspase-1 in patients with AOSD (after Bonferroni's correction).**

|  | Correlation Coefficient | P-value |
|---|---|---|
| MIP-3a/CCL20 | 0.7849 | <0.001 |
| IL-3 | 0.7169 | <0.001 |
| MIP-3β/CCL19 | 0.7029 | <0.001 |
| LIF | 0.6683 | <0.001 |
| IL-2Ra | 0.6441 | <0.001 |
| I-TAC/CXCL11 | 0.6385 | <0.001 |
| Basic FGF | 0.6251 | <0.001 |
| 6Ckine/CCL21 | 0.6174 | <0.001 |
| MIF | 0.6125 | <0.001 |
| IL-2 | 0.5865 | <0.001 |
| IL-1Ra | 0.5862 | <0.001 |
| TECK/CCL25 | 0.5852 | <0.001 |
| CCL1 | 0.5719 | <0.001 |
| M-CSF | 0.5642 | <0.001 |
| TNF-α | 0.5542 | <0.001 |
| GM-CSF | 0.5419 | <0.006 |
| IL-18 | 0.5361 | <0.012 |

AOSD: Adult-onset Still's disease, MIP: Macrophage inflammatory protein, CCL: Chemokine (C-C motif) ligand, IL: Interleukin, LIF: Leukemia inhibitory factor, I-TAC: Interferon-inducible T cell alpha chemoattractant, CXCL: Chemokine (C-X-C motif) ligand, FGF: Fibroblast growth factor, MIF: Macrophage migration inhibitory factor, TECK: Thymus expressed chemokine, M-CSF: Macrophage colony-stimulating factor, TNF: Tumor necrosis factor.

(A)

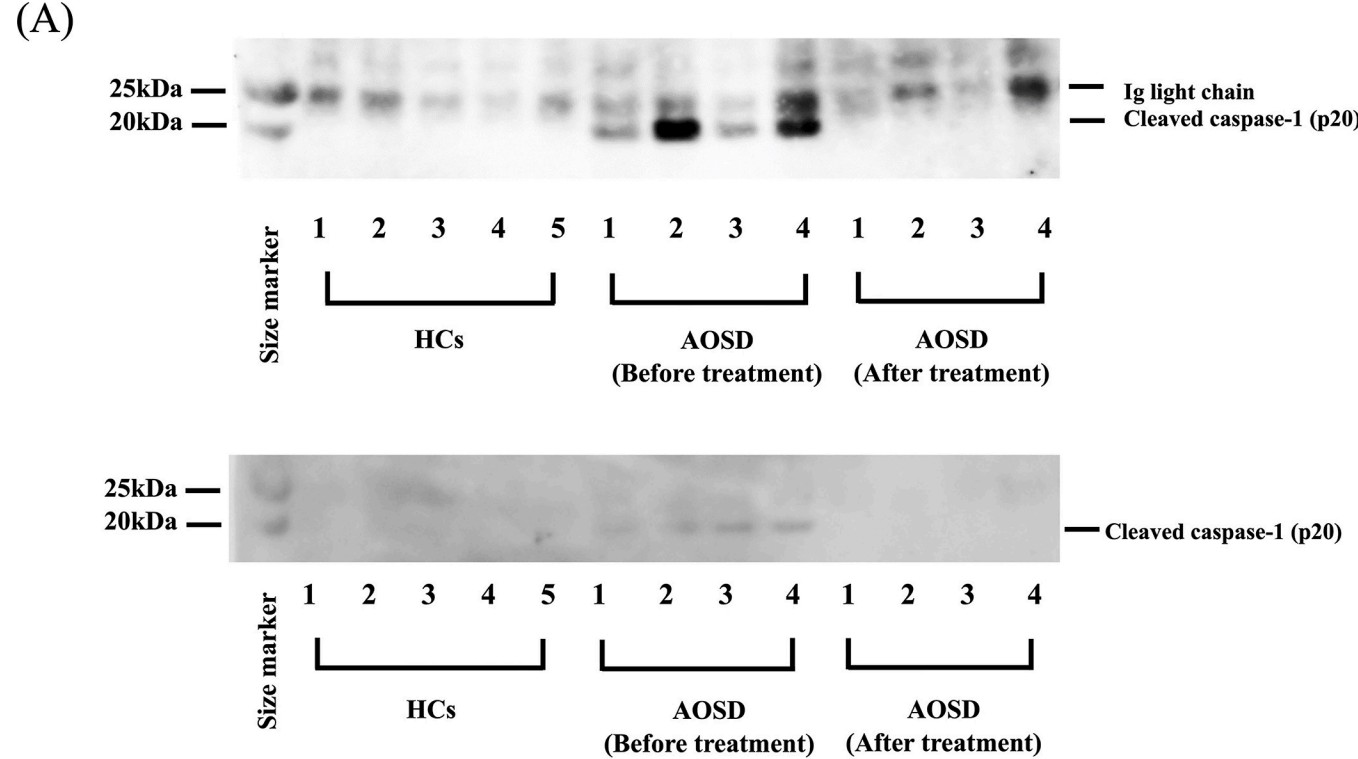

(B)

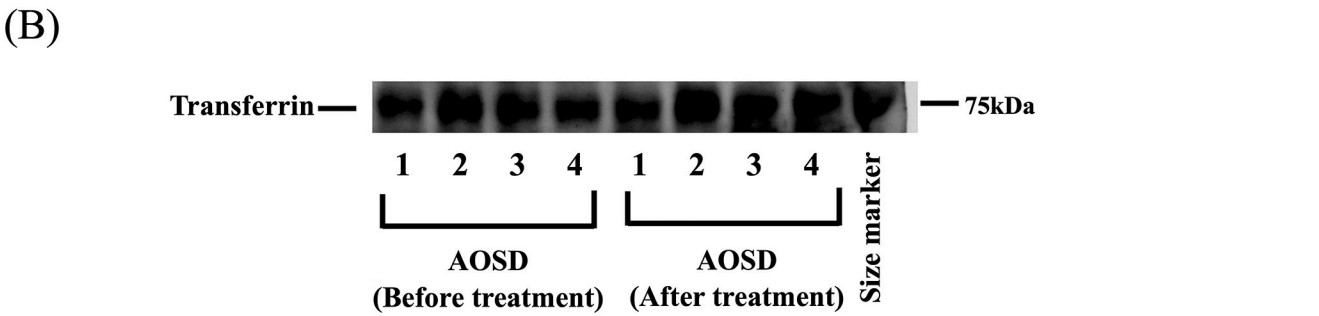

**Fig 6. Anti-caspase-1 immunoblot analysis using sera from patients with AOSD.** Sera from patients with AOSD (before or after treatments) or health HCs were dissolved in sample buffer, separated in Tris-acetate gels, and analyzed by anti-cleaved caspase-1 immunoblot analysis. (A) The cleaved caspase-1 band (p20) was detected in serum from patients with AOSD under reducing conditions. IgG light chain was detected in the upper gel, but after IgG removal by protein G beads, cleaved caspase-1 was detected as in the lower gel. (B) Serum transferrin was detected as the internal control to the same extent in untreated and treated patients with AOSD. A representative result of three independent experiments.

mediators [23]. hence, we investigated the sera of patients with AOSD for cleaved caspase-1 using immunoblot analysis. We revealed that the cleaved form of caspase-1, p20, was detectable in the serum of patients with AOSD. Active patients with AOSD had markedly increased expression of caspase-1 with its cleaved form. However, the cleaved form of caspase-1 was exclusively found in untreated patients with AOSD, and it was undetectable among patients with AOSD in the remission state receiving immunosuppressive therapy. Pyroptosis, a caspase-1-dependent type of programmed cell death, recently promoted IL-1β and IL-18 secretion during the inflammatory response [24]. Caspase-1 has activated the transcription factor NF-kB, independent of its

enzymatic activity, which may cause inflammation *in vivo* [25]. This indicates the presence of cross-talk between blood caspase-1 and the inflammatory mediators in the inflammatory disorder. Altogether, caspase-1 may have a potential role in inflammatory cascades by improving inflammatory cytokine production.

Recent studies have confirmed the direct association of IL-1β or IL-18 activation by caspase-1 with inflammatory environments [26]. Overexpressed caspase-1 has been demonstrated in the aortas of patients with coronary atherosclerosis [27]. Caspase-1 can be widely activated in endothelial cells and circulating innate immune cells in autoinflammation [28]. However, the role of caspase-1 in AOSD remains unclear. In our study, upregulated circulating caspase-1 was associated with inflammasome activation in AOSD. These results indicate that, in AOSD, serum caspase-1 reflects the activation status of circulating innate immune cells. Caspase-1-dependant inflammasome activation contributes to the autoinflammatory process in AOSD, aggravating the maturation and release of proinflammatory cytokines, IL-1β and IL-18.

Our study has several limitations. First, multivariate analysis could not be performed due to the small sample size. Additionally, the analysis included a small number of patients with AOSD. Second, multiple serum cytokines were measured in patients with AOSD only, and not in control patients (RA) or HCs. Additionally, characteristic baseline differences between patients with AOSD, those with RA, and HC differ in terms of age and gender; thus, a background-aligned study is further warranted. Third, the mechanism behind caspase-1 release was not completely elucidated. Further studies require a larger number of cases and innate immune cellular analysis, which elucidates the mechanisms for elevating circulating caspase-1.

## Conclusions

Serum caspase-1 levels, a component of the inflammasome, were elevated in patients with AOSD. Serum caspase-1 levels in patients with AOSD significantly correlated with disease activity indices, such as serum ferritin and Pouchet's score, as well as with inflammatory cytokines, including IL-18. Thus, caspase-1 may be an effective biomarker for diagnosing AOSD and predicting disease activity. However, the early diagnosis and prognostic value of caspase-1 in patients with AOSD require further validation with large-sample studies.

## Supporting information

**S1 Fig. Original gel image of Fig 6A, upper gel.** The cleaved caspase-1 band (p20) was detected in serum from patients with AOSD under reducing conditions (20 kDa). IgG light chain was also detected (25 kDa).
(TIFF)

**S2 Fig. Original gel image of Fig 6A, lower gel.** After IgG removal by protein G beads, cleaved caspase-1 was detected (20 kDa).
(TIFF)

**S3 Fig. Original gel image of Fig 6B.** Serum transferrin was detected as the internal control to the same extent in untreated and treated patients with AOSD.
(TIFF)

**S1 Table. The demographic data of patients with RA and HCs.**
(DOCX)

## Acknowledgments

We are grateful to Ms. Sachiyo Kanno for her technical assistance in this study.

## Author Contributions

**Conceptualization:** Haruki Matsumoto, Kiyoshi Migita.

**Data curation:** Haruki Matsumoto, Shuhei Yoshida, Yuya Fujita, Yuya Sumichika, Kenji Saito, Jumpei Temmoku, Tomoyuki Asano, Shuzo Sato.

**Formal analysis:** Haruki Matsumoto, Shuhei Yoshida.

**Funding acquisition:** Kiyoshi Migita.

**Investigation:** Haruki Matsumoto, Shuhei Yoshida, Tomohiro Koga, Yuya Fujita, Yuya Sumichika, Kenji Saito, Kiyoshi Migita.

**Methodology:** Haruki Matsumoto, Tomohiro Koga.

**Project administration:** Tomohiro Koga, Kiyoshi Migita.

**Supervision:** Masashi Mizokami, Masaya Sugiyama, Kiyoshi Migita.

**Validation:** Tomohiro Koga, Kiyoshi Migita.

**Visualization:** Tomohiro Koga.

**Writing – original draft:** Haruki Matsumoto, Kiyoshi Migita.

**Writing – review & editing:** Masashi Mizokami, Masaya Sugiyama, Kiyoshi Migita.

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
