## [Decision Letter · Decision Letter 0]

5 Apr 2024

PONE-D-24-06426

Increased serum caspase-1 in adult-onset Still’s disease

PLOS ONE

Dear Dr. Migita,

Thank you for submitting your manuscript to PLOS ONE. After careful consideration, we feel that it has merit but does not fully meet PLOS ONE’s publication criteria as it currently stands. Therefore, we invite you to submit a revised version of the manuscript that addresses the points raised during the review process.

In particular, multiple concerns were raised by reviewer 2. The authors are encouraged to review the comments, and appropriately address the concerns for acceptance. 

We look forward to receiving your revised manuscript.

Kind regards,

Yoshito Nishimura, MD, PhD, MPH

Academic Editor

PLOS ONE

Journal Requirements:

2. Please provide additional details regarding participant consent. In the ethics statement in the Methods and online submission information, please ensure that you have specified (1) what type of consent you obtained (for instance, written or verbal, and if verbal, how it was documented and witnessed). If your study included minors, state whether you obtained consent from parents or guardians. If the need for consent was waived by the ethics committee, please include this information.

   "The study was supported by the Japan Grant-in-Aid for Scientific Research (20K08777)."

4. In the online submission form you indicate that your data is not available for proprietary reasons and have provided a contact point for accessing this data. Please note that your current contact point is a co-author on this manuscript. According to our Data Policy, the contact point must not be an author on the manuscript and must be an institutional contact, ideally not an individual. Please revise your data statement to a non-author institutional point of contact, such as a data access or ethics committee, and send this to us via return email. Please also include contact information for the third party organization, and please include the full citation of where the data can be found.

Reviewers' comments:

Reviewer's Responses to Questions

**Comments to the Author**

1. Is the manuscript technically sound, and do the data support the conclusions?

Reviewer #1: Yes

Reviewer #2: Partly

2. Has the statistical analysis been performed appropriately and rigorously? 

Reviewer #1: Yes

Reviewer #2: No

3. Have the authors made all data underlying the findings in their manuscript fully available?

Reviewer #1: Yes

Reviewer #2: No

4. Is the manuscript presented in an intelligible fashion and written in standard English?

Reviewer #1: Yes

Reviewer #2: Yes

5. Review Comments to the Author

Reviewer #1: Thank you for the opportunity to review this article.

Some English grammar was not typically used in scientific articles. You could proofread it again before submission.

You wrote in the conclusion of the study, "Our findings revealed that serum caspase-1 was elevated in AOSD patients and that elevated caspase-1 was correlated with systemic scores and inflammatory cytokines including IL-18 in AOSD patients." This should be in the conclusion section or omitted.

In scientific articles, you might not want to explain the results in the figure caption. Instead, you could include the results in the main text.

In Figure 1 caption, what does it mean by "the four groups (p<0.001)"? Also, you could explain which statistical analysis you have used in the methods.

In Figure 3, you wrote "Japanese," it should be changed to "English."

Would you include baseline characteristics of HCs and RA patients in tables, or have you omitted them? Differences in baseline characteristics are also significant limitations.

You could explain briefly why the each limitations you suggest are important.

Reviewer #2: This study was to assess the level of caspase-1 in AOSD patients for the diagnosis and monitoring disease activity. While there are several major issues that should be take care of.

1.Are there any data about the behaviors of caspase-1 in the event of a flare (other than MAS) in patients with AOSD?

2.Is the kinetics of caspase-1 increase/decrease different from that of ferritin or CRP or ESR?

3.I have not found any information about the ongoing treatment of AOSD patient (and controls) included in the study. Besides，clinical manifestations related to Pouchot score should be provided in table 1.

4.How do the levels of caspase-1 compare between patients with MAS and non-MAS, as well as between those with active and inactive disease, considering the potential role of Caspase-1 to access disease activity.

5.The article suggests that caspase-1 can be used for diagnosing AOSD. What are the values for the area under the ROC curve (AUC), specificity, and sensitivity of caspase-1 in distinguishing between AOSD and RA?

6.Among the 71 serum inflammatory factors tested, what specific information was obtained from AOSD patients, and how many cases were included in the analysis?

7.Given the presence of numerous albumin and immunoglobulins in serum, it is necessary to remove these proteins before further testing by precipitating the proteins. Additionally, Western blot analysis for serum caspase-1 lacks internal controls.

6. PLOS authors have the option to publish the peer review history of their article (what does this mean?). If published, this will include your full peer review and any attached files.

Reviewer #1: No

Reviewer #2: No

---

## [Author Response · Author response to Decision Letter 0]

12 Jun 2024

→Thank you for your comment. We confirmed that our manuscript meets PLOS ONE`s style requirements.

2. Please provide additional details regarding participant consent. In the ethics statement in the Methods and online submission information, please ensure that you have specified (1) what type of consent you obtained (for instance, written or verbal, and if verbal, how it was documented and witnessed). If your study included minors, state whether you obtained consent from parents or guardians. If the need for consent was waived by the ethics committee, please include this information.

→Thank you for your comment. We described that “The institutional Review Board waived the need for written informed consent from participants due to the non-interventional design of the retrospective study” in part of patients and study design.

 "The study was supported by the Japan Grant-in-Aid for Scientific Research (20K08777)."

→Thank you for your comment. We added the following text to the cover letter: "The funders had no role in study design, data collection and analysis, decision to publish, or preparation of the manuscript." 

4. In the online submission form you indicate that your data is not available for proprietary reasons and have provided a contact point for accessing this data. Please note that your current contact point is a co-author on this manuscript. According to our Data Policy, the contact point must not be an author on the manuscript and must be an institutional contact, ideally not an individual. Please revise your data statement to a non-author institutional point of contact, such as a data access or ethics committee, and send this to us via return email. Please also include contact information for the third party organization, and please include the full citation of where the data can be found.

→Thank you for your comment. We revised the data statement to a non-author institutional point of contact (Contact information for the representative of our medical office managing the data set). 

→Thank you for your comment. We revised the Patients and study design part and added the following sentence: “This study was conducted in accordance with the principles of the Declaration of Helsinki and approved by the institutional review boards of Fukushima Medical University (No. 2021-290; the date of approval: 16 - April 2023) and the National Center for Global Health and Medicine (NCGM-G-003472; the date of approval: 19 - April 2023). ”

→Thank you for your comment. We added the following sentence: “We submitted gel image as the supporting information.”

Reviewers' comments:

Reviewer's Responses to Questions

Comments to the Author

Reviewer #1: Thank you for the opportunity to review this article.

Some English grammar was not typically used in scientific articles. You could proofread it again before submission.

→Thank you for your comment. We proofread the final version of the manuscript again in English.

You wrote in the conclusion of the study, "Our findings revealed that serum caspase-1 was elevated in AOSD patients and that elevated caspase-1 was correlated with systemic scores and inflammatory cytokines including IL-18 in AOSD patients." This should be in the conclusion section or omitted.

→Thank you for your comment. In the introduction, we omitted the following sentence: “Our findings revealed that serum caspase-1 was elevated in AOSD patients and that elevated caspase-1 was correlated with systemic scores and inflammatory cytokines, including IL-18 in AOSD patients.”

In scientific articles, you might not want to explain the results in the figure caption. Instead, you could include the results in the main text.

In Figure 1 caption, what does it mean by "the four groups (p<0.001)"? Also, you could explain which statistical analysis you have used in the methods.

→Thank you for your comment. “the four groups (p<0.001)" was a mistake. The correct description is shown in Figure legend as follows; The AOSD group had significantly higher caspase-1 levels for both HC and RA groups (p<0.001).

In Figure 3, you wrote "Japanese," it should be changed to "English."

→Thank you for your comment. We revised the Figure 3 you pointed out.

Would you include baseline characteristics of HCs and RA patients in tables, or have you omitted them? Differences in baseline characteristics are also significant limitations.

You could explain briefly why the each limitations you suggest are important.

→Thank you for your comment. Demographic data of HC and patients of RA were attached in Supplemental Table1 and following sentence was added to the result section; “24 of the HCs were female, and the average age was 41 years [IQR]; 28–52 years. They had no medical history or ongoing medications, and no abnormalities were noted in recent physical examinations. The demographic data of patients with RA and HCs were attached in Supplemental Table 1.” Characteristic baseline differences between AOSD patients, RA patients, and HC are described in the limitation section; “Additionally, characteristic baseline differences between AOSD patients, RA patients, and HC differ in terms of age and gender, so a background-aligned study is necessary in further.”

Reviewer #2: This study was to assess the level of caspase-1 in AOSD patients for the diagnosis and monitoring disease activity. While there are several major issues that should be take care of.

1.Are there any data about the behaviors of caspase-1 in the event of a flare (other than MAS) in patients with AOSD?

→Thank you for your comment. Data at the time of a flare other than MAS were not collected and considered in this study.

2.Is the kinetics of caspase-1 increase/decrease different from that of ferritin or CRP or ESR?

→Although ESR data are lacking, no correlation was found between CRP and Caspase-1 (r=0.143, p=0.356). The correlation with Ferritin is shown in Fig. 3. The following sentence was added to the result. “However, no correlation was found between CRP and caspase-1 (r=0.143, p=0.356) (data was not shown in figure)”

3.I have not found any information about the ongoing treatment of AOSD patient (and controls) included in the study. Besides，clinical manifestations related to Pouchot score should be provided in table 1.

→Thank you for your comment. We added the information of details of treatment and clinical manifestations related to Pouchot`s score.

4.How do the levels of caspase-1 compare between patients with MAS and non-MAS, as well as between those with active and inactive disease, considering the potential role of Caspase-1 to access disease activity.

Comparison of caspase-1 between MAS and non-MAS showed no significant difference at p=0.189. This was added in the result section as follow; “Additionally, the comparison of caspase-1 between MAS and non-MAS showed no significant difference (p=0.189).”

Since many of the active and inactive data we collected were for non-MAS (6/8 patients had non-MAS, and 2/8 patients had MAS), it was not possible to compare active and inactive data for MAS and non-MAS, respectively. 

5.The article suggests that caspase-1 can be used for diagnosing AOSD. What are the values for the area under the ROC curve (AUC), specificity, and sensitivity of caspase-1 in distinguishing between AOSD and RA?

→Thank you for your comment. We showed the values for the area under the ROC curve in Fig 2 and added the following sentence in the result section; “We analyzed the cut-off value for the caspase-1 in distinguishing between AOSD and RA by obtaining the ROC curve, which revealed 26.9 ng/mL (Sensitivity 100.0% and Specificity 93.9%). Area under the ROC curve was 0.987. (Figure 2).”

6.Among the 71 serum inflammatory factors tested, what specific information was obtained from AOSD patients, and how many cases were included in the analysis?

→Cytokines were analyzed in all 51 AOSD patients. The comprehensive analysis of cytokines was performed only in AOSD, and we did not search which cytokines were specifically elevated compared to RA and HC. However, IL-12/IL-23, IL-5, 7, and 13 were detected with sensitivity or higher in only about half of all AOSD cases, suggesting that these cytokines may have little relation to the inflammatory pathogenesis of AOSD among 71 cytokines. This point was described in the limitation section as follows; “Second, multiple serum cytokines were measured in AOSD patients only, and not in control patients (RA) or HCs.”

7.Given the presence of numerous albumin and immunoglobulins in serum, it is necessary to remove these proteins before further testing by precipitating the proteins. Additionally, Western blot analysis for serum caspase-1 lacks internal controls.

→Thank you for your comment. As you pointed out, fragmentation of the serum IgG into IgG light chains (25 kDa, respectively) was detected by the secondary antibody used for western blot. To avoid these interfering IgG-derived bands, sera were pre-absorbed using protein G beads and subjected anti-cleaved caspase-1 western blot. As shown in Figure 6B, single cleaved caspase-1 bands (20kDa) were exclusively detected in the sera from untreated AOSD patients. Additionally, as internal controls, we confirmed that serum transferrin was detected to the same extent in untreated and treated AOSD patients (Figure 6C). These results were added in the results section.

---

## [Decision Letter · Decision Letter 1]

15 Jul 2024

Increased serum caspase-1 in adult-onset Still’s disease

PONE-D-24-06426R1

Dear Dr. Migita,

We’re pleased to inform you that your manuscript has been judged scientifically suitable for publication and will be formally accepted for publication once it meets all outstanding technical requirements.

Kind regards,

Yoshito Nishimura, MD, PhD, MPH

Academic Editor

PLOS ONE

Additional Editor Comments (optional):

Reviewers' comments:

Reviewer's Responses to Questions

**Comments to the Author**

1. If the authors have adequately addressed your comments raised in a previous round of review and you feel that this manuscript is now acceptable for publication, you may indicate that here to bypass the “Comments to the Author” section, enter your conflict of interest statement in the “Confidential to Editor” section, and submit your "Accept" recommendation.

Reviewer #1: All comments have been addressed

Reviewer #3: All comments have been addressed

Reviewer #4: All comments have been addressed

2. Is the manuscript technically sound, and do the data support the conclusions?

Reviewer #1: Yes

Reviewer #3: Yes

Reviewer #4: Yes

3. Has the statistical analysis been performed appropriately and rigorously? 

Reviewer #1: Yes

Reviewer #3: Yes

Reviewer #4: Yes

4. Have the authors made all data underlying the findings in their manuscript fully available?

Reviewer #1: Yes

Reviewer #3: Yes

Reviewer #4: Yes

5. Is the manuscript presented in an intelligible fashion and written in standard English?

Reviewer #1: Yes

Reviewer #3: Yes

Reviewer #4: Yes

6. Review Comments to the Author

Reviewer #1: Thank you very much for the opportunities reviewing your manuscript all issues I pointed out seems to be addressed.

Reviewer #3: All my queries were addressed. I have no further comments on the manuscript. In this form, manuscript can be published.

Reviewer #4: In this study, the authors evaluated the potential of serum caspase-1 levels as a useful inflammatory biomarker in patients with adult-onset Still's disease (AOSD). They measured serum caspase-1 levels in 51 consecutive patients diagnosed with AOSD based on Yamaguchi criteria using enzyme-linked immunosorbent assay and compared them with controls, rheumatoid arthritis (RA) patients, and healthy controls (HC). The results showed that serum caspase-1 levels were significantly elevated in AOSD patients than in RA and HC patients. Serum caspase-1 showed a significant positive correlation with disease activity score, serum ferritin level, and the levels of inflammatory cytokines such as IL-18. Immunoblot analysis detected a truncated form of caspase-1 (p20) in sera from untreated AOSD patients but not from inactive AOSD patients receiving immunosuppressive treatment. They also performed a cluster analysis of serum cytokines in AOSD patients. From the above, the authors conclude that caspase-1 is a useful biomarker for the diagnosis and monitoring of AOSD. The authors appropriately addressed the reviewer's comments and revised the paper.

7. PLOS authors have the option to publish the peer review history of their article (what does this mean?). If published, this will include your full peer review and any attached files.

Reviewer #1: No

Reviewer #3: No

Reviewer #4: No

---

## [Editor Report · Acceptance letter]

19 Jul 2024

PONE-D-24-06426R1 

PLOS ONE

Dear Dr. Migita, 

I'm pleased to inform you that your manuscript has been deemed suitable for publication in PLOS ONE. Congratulations! Your manuscript is now being handed over to our production team.

Kind regards, 

on behalf of

Dr. Yoshito Nishimura 

Academic Editor

PLOS ONE